# Noninvasive Cardiac Imaging in Formerly Preeclamptic Women for Early Detection of Subclinical Myocardial Abnormalities: A 2022 Update

**DOI:** 10.3390/biom12030415

**Published:** 2022-03-07

**Authors:** Yentl Brandt, Chahinda Ghossein-Doha, Suzanne C. Gerretsen, Marc E. A. Spaanderman, M. Eline Kooi

**Affiliations:** 1Department of Radiology and Nuclear Medicine, Maastricht University Medical Centre, 6229 HX Maastricht, The Netherlands; y.brandt@maastrichtuniversity.nl (Y.B.); s.gerretsen@mumc.nl (S.C.G.); 2CARIM School for Cardiovascular Diseases, Maastricht University, 6229 ER Maastricht, The Netherlands; chahinda.ghossein@mumc.nl; 3Department of Obstetrics and Gynecology, Maastricht University Medical Centre, 6229 HX Maastricht, The Netherlands; marc.spaanderman@mumc.nl; 4Department of Cardiology, Maastricht University Medical Centre, 6229 HX Maastricht, The Netherlands; 5GROW School for Oncology and Developmental Biology, Maastricht University, 6229 ER Maastricht, The Netherlands; 6Department of Obstetrics and Gynecology, Radboud University Medical Centre, 6525 GA Nijmegen, The Netherlands

**Keywords:** preeclampsia, cardiac imaging, cardiovascular, cardiac ultrasound, CMR, cardiac strain, tissue mapping, echocardiography

## Abstract

Preeclampsia is a maternal hypertensive disease, complicating 2–8% of all pregnancies. It has been linked to a 2–7-fold increased risk for the development of cardiovascular disease, including heart failure, later in life. A total of 40% of formerly preeclamptic women develop preclinical heart failure, which may further deteriorate into clinical heart failure. Noninvasive cardiac imaging could assist in the early detection of myocardial abnormalities, especially in the preclinical stage, when these changes are likely to be reversible. Moreover, imaging studies can improve our insights into the relationship between preeclampsia and heart failure and can be used for monitoring. Cardiac ultrasound is used to assess quantitative changes, including the left ventricular cavity volume and wall thickness, myocardial mass, systolic and diastolic function, and strain. Cardiac magnetic resonance imaging may be of additional diagnostic value to assess diffuse and focal fibrosis and perfusion. After preeclampsia, sustained elevated myocardial mass along with reduced myocardial circumferential and longitudinal strain and decreased diastolic function is reported. These findings are consistent with the early phases of heart failure, referred to as preclinical (asymptomatic) or B-stage heart failure. In this review, we will provide an up-to-date overview of the potential of cardiac magnetic resonance imaging and echocardiography in identifying formerly preeclamptic women who are at high risk for developing heart failure. The potential contribution to early cardiac screening of women with a history of preeclampsia and the pros and cons of these imaging modalities are outlined. Finally, recommendations for future research are presented.

## 1. Introduction

Preeclampsia is a hypertensive maternal syndrome, complicating 2–8% of pregnancies, with short-term but also long-term health effects in these women. Preeclampsia is thought to be related to endothelial dysfunction, superimposed on preexisting circulatory, metabolic, hemostatic, and immunological abnormalities [1,2]. Women who had preeclampsia have a 2- to 7-fold higher probability of developing cardiovascular diseases (CVDs) later in life. These include, among others, ischemic heart disease, cerebrovascular accidents, cardiac arrhythmias, heart failure, and diastolic dysfunction [2]. A recent study by Vogel et al. estimated the worldwide prevalence of cardiovascular disease in women to be 6403 cases per 100,000 [3]. According to a meta-analysis performed by Wu et al., preeclampsia is associated with a 4-fold increase in heart failure (risk ratio (RR), 4.19; 95% confidence interval (CI), 2.09–8.38), and a 2-fold increase in coronary heart disease and stroke (RR, 2.21; 95% CI, 1.83–2.66 and RR, 1.81; 95% CI, 1,29–1.55, respectively) [4].

Several key processes are considered to play a vital role in the cardiac sequelae after preeclampsia. On the one hand, prior preeclampsia is associated with structural and functional vascular changes, as well as cardiac remodeling of the left ventricle that may predispose to microcirculatory shortfall [5]. This remodeling is thought to be a result of cardiac adaptation to an increase in total peripheral vascular resistance during pregnancy from the 29th week onward, which occurs alongside an increase in blood pressure [6,7,8]. On the other hand, macrovascular coronary atherosclerosis is also considered to play a role since formerly preeclamptic women develop coronary artery calcifications on average 5 years earlier than women who underwent a normotensive pregnancy [9].

In 25% to 72% of the cases, these cardiac adaptations persist and do not revert to a pre-pregnancy state postpartum, and a majority of women meet the diagnostic criteria for asymptomatic heart failure preterm. This gives formerly preeclamptic women a higher vulnerability to develop cardiovascular disease later in life [2,10].

Our current understanding of the pathophysiological transition from healthy, to asymptomatic heart failure, to evident clinical heart failure in the post-preeclampsia aftermath is limited. Therefore, in recent years several studies have attempted to elucidate this transition, as early identification and subsequent treatment of subclinical heart failure may improve treatment outcomes and reduce hospitalization rate. Despite the predisposition of former preeclamptic women to CVD, current screening methods are still insufficient to properly identify women at risk of developing CVD [11,12]. A limited number of studies investigated cardiovascular health of formerly preeclamptic women between the ages of 40 to 60 years when early signs of CVD are most likely to be revealed [12]. In this period, left ventricular remodeling, as well as capillary rarefaction (a hallmark of microvascular dysfunction), become evident in a significant proportion of women after hypertensive pregnancy [13]. This supports the view that cardiac dysfunction may not only originate from macrovascular disease but also from microvascular dysfunction.

Cardiac imaging studies to investigate structural and functional cardiac abnormalities in formerly preeclamptic women primarily focused on the use of cardiac ultrasound and, to a lesser extent, on cardiac magnetic resonance imaging (CMR) [13,14,15,16,17,18].

This review article will provide an up-to-date overview of the current state-of-the-art applications of noninvasive cardiac imaging in the early detection of post-preeclampsia cardiac changes. The pros and cons of the various cardiac imaging methods will be discussed. The results of current research will be summarized. Finally, recommendations for future directions will be given.

## 2. Cardiac Magnetic Resonance Imaging

Cardiac magnetic resonance imaging (CMR) is the application of MRI to visualize the heart. CMR can be used to obtain anatomical information of the heart and myocardium as well as functional information of myocardium, perfusion, and heart valves [19]. By positioning a subject in the high magnetic field of the MRI system, protons will align with the direction of the magnetic field of the external magnet within the MRI scanner (the B_0_ field), creating a net magnetization in the direction of the B_0_ field. By manipulating this net magnetization through radiofrequency (RF) excitation pulses, the direction of the net magnetization can temporarily be perturbed. Immediately after an excitation pulse, the net magnetization in the longitudinal direction decreases, while the net magnetization in the transverse plane increases, after which it will return to its equilibrium state in the direction of the B_0_ field, also known as relaxation. This relaxation occurs at characteristic relaxation times for different tissues, and these differences are explored to generate contrast between different tissues and fluids. Differences in longitudinal relaxation are used to generate T_1_-weighted images or T_1_ maps, whereas differences in transverse relaxation are used to generate T_2_-weighted images or T_2_ maps.

One of the key hurdles in CMR is compensating for breathing and cardiac motion. Commonly used techniques to compensate for breathing motion and cardiac motion are timed breathholds, one- or two-dimensional navigators (where breathing motion is corrected for or image acquisition is limited to the end expiration phase, thus allowing free breathing), and cardiac triggering and gating (where images are acquired in a specific time window of the cardiac cycle, determined by vectorcardiography) [20].

The application of magnetic resonance imaging is often complicated by high costs and lower availability, most notably in developing countries. In many such cases, cardiac ultrasonography remains the modality of choice due to ease of use, lower costs, and more widespread availability.

In the following paragraphs, we will discuss the techniques that could be applied to examine myocardial anatomical and functional changes in formerly preeclamptic women.

### 2.1. Cine MRI

Imaging of the heart for the purpose of inspecting the volumes and geometry, as well as myocardial and valvular function, is performed with cine MRI, where a series of images is acquired in different cardiac phases in a certain orientation (plane) (Figure 1).

Assessment of the left ventricular volume and geometry (wall thickness) is typically performed using the left ventricular short-axis plane. By contouring the epicardium and the endocardium in each slice in the end-systolic and end-diastolic phase, the end-systolic and end-diastolic volume of the left ventricular cavity can be quantified. The stroke volume is the difference between the end-diastolic and end-systolic volume. From the stroke volume, the left ventricular ejection fraction (LVEF) can be calculated:(1)$$LVEF=\frac{EDV-ESV}{EDV}\times 100\%=\frac{SV}{EDV}\times 100\%$$
where *LVEF* = left ventricular ejection fraction (%), *EDV* = end-diastolic volume (mL), *ESV* = end-systolic volume (mL). Note that the difference between *EDV* and *ESV* is equal to the stroke volume (*SV*).

By measuring the area between the epicardium and endocardium in consecutive slices, the left ventricular mass (LVM) can also be determined by multiplying the volume of the left ventricular wall with the gravitational constant of the myocardium. These values are often indexed for body surface area to adjust for body size [22,23].

CMR is currently considered to be the gold standard for the measurement of cardiac volumes and geometry due to its ability to image the entire left ventricle without relying on further modeling. Superior soft-tissue contrast and an adequate temporal resolution are other advantages of cine MRI [24].

### 2.2. Myocardial Fibrosis

Myocardial fibrosis is an increase in the formation of collagen deposits in the myocardium. Fibrosis can occur focally, for example, scar tissue resulting from an old myocardial infarction, or can be diffuse, in which case these deposits are not restricted to a single area. Focal fibrosis often leads to a locally diminished systolic and diastolic function, whereas diffuse fibrosis often affects the entire myocardium, most notably by stiffening the myocardium and reducing myocardial compliance, resulting in loss of diastolic function [25,26].

Focal fibrosis is assessed by late gadolinium enhancement (LGE) [27]. Approximately 10 min after the injection of a gadolinium-based contrast medium, through either a single or double dose of 0.1–0.2 mmol/kg bodyweight, T_1_-weighted inversion recovery gradient echo imaging is applied to show possible areas of focal hyperintensity, a direct measure for focal fibrosis [28,29]. Dark blood LGE, wherein the signal of the blood pool is suppressed, thus rendering it dark or black, increases the contrast between focal fibrosis and the blood pool [27]. This technique has been validated to detect focal fibrosis with excellent results when compared to histology, the current gold standard [30].

Diffuse fibrosis is assessed through the calculation of the extracellular volume (ECV). This method has been validated extensively by comparison with histology, with an intraclass correlation coefficient of 0.884 (95% CI: 0.854, 0.914) [31,32]. By acquiring native (before contrast injection) and post-contrast T_1_ maps (approximately 10–15 after injection of a gadolinium-based contrast medium) [30], the T_1_ relaxation times for the myocardium and blood pool can be determined in both the native and post-contrast state. In addition to native and post-contrast T_1_ relaxation times, timely (within 24 h) measurement of hematocrit is also needed for the final calculation of the ECV. Alternatively, also the native T_1_ relaxation times are used as a measure for diffuse fibrosis, but this value is more dependent on various parameters, such as the MRI pulse sequence and field strength. Standard practice is to measure the T_1_ relaxation times in the septal region of a mid-height left ventricular short-axis (LVSAX) slice, though global measurements are sometimes also performed by contouring the entire myocardium (Figure 2) [30].

T_1_ relaxation times are highly dependent on field strength, with lower values reported at 1.5 T as opposed to 3.0 T. Common cardiac MRI sequences for the measurement of the T_1_ relaxation times of the myocardium include modified look-locker inversion (MOLLI), shortened MOLLI (shMOLLI), saturation recovery single-shot acquisition (SASHA), and saturation pulse prepared heart rate-independent inversion recovery (SAPPHIRE) [33,34]. While MOLLI and shMOLLI were reported to have superior precision ((4.0 (MOLLI) and 5.6 ms (shMOLLI) as compared to 8.7 ms (SASHA) and 6.8 ms (SAPPHIRE), accuracy was reported to be inferior (62 ms (shMOLLI) and 44 ms (MOLLI) as compared to 13 ms (SASHA) and 12 ms (SAPPHIRE)) [34]. The reproducibility was similar among all sequences [34]. The ECV is calculated with the following formula [35]:(2)$$ECV\;=\;\left(1-HTC\right)\frac{\frac{1}{Post-Contrast\;T_{1}\;\left(Myocardium\right)}-\frac{1}{Native\;T_{1}\;\left(Myocardium\right)}}{\frac{1}{Post-Contrast\;T_{1}\;\left(Blood\right)}-\frac{1}{Native\;T_{1}\;\left(Blood\right)}}$$
where *ECV* = extracellular volume (%), *HTC* = hematocrit (%), *T*_1_ = *T*_1_ relaxation time (ms).

### 2.3. Myocardial Strain

Whereas the left ventricular ejection fraction can be used as an indicator of systolic function, it gives little insight into the actual contraction and relaxation patterns. Myocardial strain can be used to determine the actual cardiac contractibility and relaxation in terms of the maximal relative cardiac deformation over the cardiac cycle (peak strain), as well as the maximal rate at which it deforms (peak strain rate). These deformations occur in three principal directions (radial, circumferential, and longitudinal) (Figure 3) [36].

In certain circumstances, such as preclinical left ventricular dysfunction, the LVEF may not yet be affected, but left ventricular contractability as quantified by strain assessment can already be diminished [37].

There currently exist two cardiac MRI methods for the assessment of myocardial strain: feature tracking and tissue tagging [38]. CMR-tagging is considered as the benchmark for the validation of other myocardial strain techniques and is thus the current gold standard [39,40], though it has not yet been introduced into standard clinical practice and screening [30]. Feature tracking operates on the principles of left ventricular wall delineation in combination with the identification of myocardial features identified within the left ventricular wall and the subsequent deformation of these features over the cardiac cycle, with a stronger weighting of the deformation of features closest to the endocardial border specifically, which may explain a small bias when compared to the gold standard; tagging MRI (per example; tagging vs. feature tracking: −8.0% mean difference, *p* = 0.005 at 1.5T and −6.3% mean difference, *p* < 0.001 at 3.0T for global circumferential strain [37,41]. Figure 4 exemplifies the principal strain directions in measurement.

Rahman et al. published an extensive systematic review article in which they found a reasonable to decent agreement for feature tracking when compared to 2D echo-based speckle tracking (9.5% and 16.4% inter-modality variation for global circumferential strain and global longitudinal strain, respectively) [42]. Feature tracking can be applied using the same cine MR images that are acquired for the assessment of cardiac geometry and volumes (LV4CH and LV2CH for longitudinal strain, LVSAX for radial and circumferential strain). Tissue tagging, on the other hand, is based on the principle of creating a magnetic overlay during image acquisition, where locally induced disturbances of the net magnetization with specific radiofrequency pre-pulses create dark lines over the anatomical image, which can be applied in two orthogonal planes, creating a grid-like appearance where tags become visible in the LVSAX plane [38]. These are dot-like structures that indicate the location of tissue. Because the position of these tags depends on the tissue magnetization itself, they will deform over the course of the cardiac cycle, and the deformation of these tags can then be quantified to allow for a direct assessment of myocardial deformation, and thus, myocardial radial and circumferential strain [38]. Compared to tagging, feature tracking often overestimates strain values (longitudinal peak strain; −11.05% ± 3.10% vs. −16.47% ± 3.07% for tagging and feature tracking, respectively) [43]. Interstudy reproducibility for feature tracking-based peak strain is considered excellent for feature tracking (coefficient of variation of 6.4% and 11.2% for 1.5 T and 3.0 T, respectively), and suitable for tagging (coefficient of variation of 16.7% and 14.4% for 1.5 T and 3.0 T, respectively). For systolic strain rates, the reproducibility is only suitable (coefficient of variation of 15.1% and 23.0% for tagging at 1.5 T and 3.0 T, respectively, and 13.1% and 18.6% for feature tracking at 1.5 T and 3.0 T, respectively) [43]. Given the propensity for individual tags to fade at the end of the cardiac cycle, feature tracking is preferred to assess diastolic strain rates.

### 2.4. Myocardial Perfusion

Women with a history of preeclampsia are at increased risk of developing heart failure later in life, either with or without preserved ejection fraction [44,45]. The more prevalent heart failure with preserved ejection fraction in women as compared to men is thought to originate from myocardial microvascular dysfunction [46].

Myocardial microvascular dysfunction can be assessed through first-pass perfusion MRI (Figure 5). A bolus of contrast medium is injected during dynamic short-axis imaging of the left ventricle. This will first lead to a steep increase in signal intensity of the left ventricle cavity, which is followed by a less steep increase in signal intensity of the left ventricular wall. Qualitative assessment remains the most used method of analyzing perfusion MRI, where regions of hyper- or hypo-intensity are visually observed. However, this method does not allow the detection of global perfusion dysfunction, and it does not provide quantitative results. Alternatively, Fermi deconvolution or other models can be employed to determine the tissue response function for quantitative assessment of the tissue perfusion in milliliters per gram of tissue per minute [45]. Another option is to derive semi-quantitative perfusion parameters, such as the ratio of the upslope of the signal intensity of the myocardium and ventricular cavity (relative upslope) [46]. By analyzing these perfusion parameters in both rest and stress (induced pharmacologically with a vasodilator, such as adenosine), the myocardial perfusion reserve index can be calculated. The myocardial perfusion reserve index is a measure of the capacity to adjust myocardial perfusion to stress stimuli [45].

### 2.5. D Flow Measurements

Diastolic function can be measured using 4D flow CMR. Using phase-contrast CMR in all three spatial dimensions, with time being the fourth, flow velocities can be assessed accurately. Furthermore, patterns of flow can be identified and classified as laminar or turbulent, among others. Flow-based diastolic parameters will be further elaborated upon in the paragraph on transthoracic cardiac ultrasonography. The clinical application of this technique is currently not yet widespread, with the majority of clinical examinations using a Doppler ultrasound-based approach [47].

## 3. Transthoracic Cardiac Ultrasonography

As compared to other visualization techniques, cardiac ultrasonography is currently the most dominant imaging modality in women with a history of preeclampsia [48]. The clinical applications of ultrasonography commonly use frequency ranges of 1.5 to 7.5 MHz [48]. By using a transducer that is both a transmitter and receiver of pulsed ultrasound waves, images can be reconstructed. As the sound wave travels through different tissues, it will encounter tissue borders where the signal will be reflected, resulting in an echo of the sound wave that returns in the opposite direction of the initial sound wave. This echo will be received by the piezoelectric crystal in the ultrasound transducer. The difference in time between the emission of the sound wave and the detection of its subsequent echo can be used to determine tissue depth based on the speed of sound within tissue. This can be in the form of a reflection, or scattering may occur if the structure has a size smaller than the wavelength of the sound wave.

### 3.1. Volumes and Function

The assessment of cardiac volumes and geometry in echocardiography follows the same principles as CMR-derived cardiac volumes and geometry. The imaging planes used in cardiac ultrasound are thus very comparable, and the measurement through delineation of the myocardial borders follows a similar process, with the caveat that echocardiography is typically performed in one single imaging plane or tissue depth and is thus dependent on the use of a model for the calculation of the left ventricular volume and function [48,49].

As with CMR-derived cardiac volumes and geometry, the outcome measures of cardiac ultrasound primarily focus on the calculation of ventricular volumes. The measurement of LVEDV and LVESV allows, just as for CMR, the calculation of the LVSV, and finally the LVEF. Ultrasonography-based calculation of the LVM is performed through the use of the Devereux formula, which is then commonly indexed to the body surface area.
$$LVM=0.8\times 1.04\times [\left(IVS_{d}+LVID_{d}+PWT_{d}{)}^{3}-LVID_{d}^{3}\right]\;+\;0.6$$
where *LVM* = left ventricular mass, *IVS_d_* = ventricular septal thickness at end-diastole, *LVID_d_* = left ventricular internal diameter at end-diastole, *PWT_d_* = inferolateral (posterior) wall thickness at end-diastole.

### 3.2. Myocardial Strain

When assessing myocardial strain through cardiac ultrasonography, there are subtle differences in technique when compared to CMR-derived feature tracking. Ultrasound-derived strain measurements hinge on the technique of speckle tracking, where cardiac motion is followed during the cardiac cycle by tracking image inhomogeneities. These inhomogeneities in image texture are called speckles. More precisely, it is the speckled pattern that naturally occurs in cardiac ultrasound, which moves along with systolic and diastolic motion and is tracked using a common block-matching method, automatically identifying a speckle within an area of the contoured delineation and then comparing that feature in each temporal cardiac phase with the best match. Speckle tracking is usually performed in the long-axis (LV2CH, LV3CH, and LV4CH) view to quantify longitudinal strain. The possibility exists to employ it also to derive radial and circumferential strain, though the results of radial strain have a low reproducibility (exemplified by a high spread, −20.9% to −27.8% for circumferential strain, −15.9% to −22.1% for longitudinal strain, and 35.1% to 59.0% for radial strain) and are difficult to standardize [50].

### 3.3. Myocardial Perfusion

Apart from a CMR-based approach, myocardial perfusion can also be assessed qualitatively and quantitatively through the use of contrast-enhanced cardiac ultrasound measurements. The principles of this modality rely on the scattering of soundwaves induced by the presence of microbubbles. By calculating the mean microbubble velocity and the peak myocardial signal intensity, myocardial perfusion can be quantified [51].

### 3.4. Diastolic Function

Echocardiography can also be used to measure the velocity of blood within the heart with high temporal resolution through the Doppler effect. By applying the principles of redshift (elongation of sound frequency due to objects moving away from the probe) and blueshift (shortening of sound frequency due to objects moving toward the probe), the velocity of moving blood can be determined through the echo of erythrocytes in the blood. Different techniques for the measurement of these velocities exist. Continuous-wave (CW) Doppler ultrasonography measures velocities in the entire line of the ultrasound beam. This allows for the measurement of high velocities, but CW Doppler is unable to discern the precise location of the highest velocity. Pulse-wave (PW) Doppler measures velocities at a specific depth but is unable to accurately measure high velocities. Color-flow mapping (CFM) allows for the creation of a heatmap overlay over the anatomical image, where red areas depict regions with blood flow away from the probe, while blue areas represent flow toward it [52]. Tissue Doppler imaging (TDI) follows the same principles, but instead of the tracking of the velocity of blood, it follows the movement of the myocardium itself during the cardiac cycle [53].

The flow parameters that are usually derived from these measurements are the early diastolic inflow velocity (E), the late diastolic inflow velocity (A), systolic annular velocity (s’), early diastolic annular velocity (e’), and late diastolic annular velocity (a’) [54,55]. Calculated parameters from these measurements are the E/A ratio, which provides a ratio of ventricular filling through the mitral valve during early diastole, where ventricular compliance is the principal instigator (E), and late diastole, when the atrial contraction occurs (A). The E/e’ ratio is used to provide a ratio of the early diastolic inflow (E) and the early diastolic annular velocity (e’) [55].

## 4. Pros and Cons of CMR and Cardiac Ultrasonography

Table 1 summarizes the pros and cons of CMR as opposed to cardiac ultrasonography [56]. While CMR has superior soft-tissue contrast, it suffers from lower temporal resolution, which can hamper the detection of strain rate and diastolic function. Availability and costs also contribute to the decision of which modality to use since MRI is more expensive than ultrasonography. Still, cine MRI remains the gold standard for volume and function measurement, and tagging CMR remains the gold standard for myocardial circumferential and radial strain.

## 5. Cardiac Imaging Studies in Formerly Preeclamptic Women

### 5.1. Search Strategy

PubMed was assessed for the purpose of collecting research data for inclusion in this review. An initial search was performed with the keywords ‘cardiac imaging’, ‘postpartum imaging’, ‘MRI’, ‘ultrasound’, ‘tissue mapping’, ‘cardiac strain’, ‘echocardiography’, ‘CMR’, ‘magnetic resonance imaging’, ‘cardiac magnetic resonance,’ and ‘cardiac function’, all in combination with the term ‘preeclampsia’. The search results were then vetted for eligibility on the following points.
A clear focus on maternal postpartum cardiac imaging;Specific results of a post-preeclampsia case group;Cardiac imaging studies with the exception of imaging of the coronary arteries;Inclusion of a control group or comparing results to reported normal values in literature;Publication date <20 years;Animal studies were excluded.


Titles, abstracts, and keywords were examined for inclusion eligibility. After this initial selection, duplicates were removed. Finally, the full text of the prospective inclusion was studied for eligibility and included when deemed in line with the scope of this review. Upon inclusion, references cited by included records were screened for further inclusion according to the same principles.

A total of 1900 records were identified through the literature search strategy. The titles, abstracts, and keywords were screened for relevance, after which 62 records remained. After removing duplicates and unsystematic reviews, 28 articles remained. Finally, articles cited by the included articles were screened, providing three additional articles for a total of 31 inclusions.

### 5.2. Research Findings

The results of these studies are summarized in Table 2. The majority of studies that focus on cardiac sequelae of preeclampsia have focused on the use of ultrasonography in the detection of ventricular volume and function. The use of CMR, while the gold standard, is often not the primary imaging modality. Furthermore, the range in the postpartum interval is wide, from 1–3 days to 40 years. The mean age of the participants in most studies is around 30–45 years.

In normotensive pregnancy, ventricular wall thickening occurs during the pregnancy to compensate for the increased volumetric load. However, in contrast to hypertrophic conditions, the relative wall thickness, which is the wall thickness when adjusted for increases in volume, does not increase. Increases in ventricular wall thickness recover to a pre-pregnancy state postpartum [58]. During a hypertensive pregnancy, the relative wall thickness increases, indicating that the ventricular wall thickens disproportionately to the increase in ventricular volume, and this does not fully resolve to a pre-pregnancy state postpartum [59].

In 2019 27 echocardiography studies on cardiac function in women with a history of preeclampsia were summarized in a systematic review [60]. No significant differences between preeclamptic and non-preeclamptic women were reported for left ventricular ejection fraction, isovolumetric relaxation time, or deceleration time.

A persistent increase in left ventricular mass has been reported by some studies [7,61,62,63,64,65,66,67,68]. In contrast, others reported either no increase in left ventricular mass or a reversion to a post-pregnancy state [10,14,15,16,28,69,70,71,72,73,74,75,76,77,78,79,80,81,82,83]. The following limitations were mentioned by the authors of the studies that found no differences: a limited sample size [14,69,70,73,74,75,78,80,83], differences in the postpartum interval [71], antihypertensive medication use in the case group [77], selection bias toward patients in poorer health [78], and fetal growth restriction in the control group [80]. A systematic review demonstrated a higher left ventricular mass index in formerly preeclamptic women with a mean difference of 4.25 g/m^2^ (95% CI, 2.08, 6.42) [60]. Ghossein-Doha et al. reported an increased prevalence of cardiac remodeling 4–10 years after the preeclamptic pregnancy (1 (2%) vs. 27 (25%) for controls and formerly preeclamptic women, respectively) in a study of 107 formerly preeclamptic women and 41 controls [77]. In contrast, no significant difference in this parameter was reported by Birukov et al. in a smaller study with 22 women with a history of preeclampsia and 22 controls [16]. Inconsistent findings were also reported for relative wall thickness. Some studies reported an increase in relative wall thickness [7,65,67,68,79], while other studies reported no significant differences [64,69,70,75,77,80,83]. The following limitations were described by the authors of the studies that found no differences: again a limited sample size [64,69,70,75,80,83], antihypertensive medication use in the case group [77], and fetal growth restriction in the control group [80]. The systematic review demonstrated a higher relative wall thickness in formerly preeclamptic women with a mean difference of 0.03 cm (95% CI, 0.01, 0.05) [60]. Increases in left ventricular wall thickness, either septal or posterior, were described by several groups [61,62,65,66,67]. Other studies did not find a significant difference [69,73,75,77,80] or a significant increase immediately postpartum, which reverts to a pre-pregnancy state [70]. Limitations cited by the authors of the studies that found no differences were a limited sample size [64,69,75,80], differences in postpartum interval [71], and fetal growth restriction in the control group [80].

The vast majority of papers found a reduction in global strain in all principal directions (radial, circumferential, longitudinal) [16,62,66,73,84,85], whereas few studies found no significant difference [69,80]. The latter studies had a relatively low sample size, with one control group consisting of women whose fetuses were growth restricted [80]. Thus, subtle contractional dysfunction can already occur without loss of ejection fraction in formerly preeclamptic women, which makes global stain a promising parameter for early detection of cardiac changes.

The systematic review showed a lower E/A and a higher E/e ratio with a mean difference of −0.08 (95% CI, −0.15, −0.01) and 0.84 (95% CI, 0.41, 1.27), respectively in women with a history of preeclampsia [60].

To summarize, cardiac geometry alterations among increased left ventricular mass, relative wall thickness and absolute wall thickness, and loss in diastolic function are frequently reported as cardiac sequelae of preeclampsia. The vast majority of the studies have demonstrated a reduction in global cardiac strain, which may be a sensitive parameter to detect subtle changes in cardiac function.

Table 2 presents an overview of cardiac imaging studies in formerly preeclamptic women.

**Table 2 biomolecules-12-00415-t002:** Research findings of the in-depth literature study. PPI = postpartum interval in months (unless specified otherwise), US = ultrasound, CMR = cardiac magnetic resonance imaging, PE = preeclampsia.

Reference	Subjects (n) (Controls/PE)	PPI (Months) (Controls/PE)	Age (Years) (Controls/PE)	Study Aim	Imaging Modalities Used	Main Outcomes
Al-Nashi et al., 2016 [69]	16/15	134 ± 7/134 ± 7	41.2 ± 3.2/39.4 ± 3.6	Assessment of ventricular structure and function in formerly preeclamptic women	US	No significant remaining alterations in ventricular structure and function, myocardial strain, and diastolic function
Ambrožič et al., 2021 [70]	15/25	1 day, 12 months/1 day, 12 months	36 (31–39)/30 (27–37)	Assessment of ventricular structure and function from immediately post delivery to one year postpartum	US	Left ventricular mass was increased immediately post delivery in the PE group (125 (119–140) g vs. 152 (120–198) g, *p* = 0.003), but resolved one year postpartum Septal wall thickness was increased in the PE group (0.8 (0.8–0.9) cm vs. 0.9 (0.8–1.0) cm, *p* = 0.016), but resolved one year postpartum E/e’ was increased in the PE group (6.9 (6.4–7.8) vs. 8.7 (7.6–10.0), *p* = 0.003) immediately post delivery, but resolved one year postpartum
Birukov et al., 2020 [16]	22/22	48 ± 63/24 ± 12	Not specified, no significant difference	Assessment of the possibility of early risk stratification with CMR	CMR	No significant remaining alterations in ventricular structure and function were reported Global radial and circumferential strain were diminished in the PE group (mean difference: 4.56% ± 2.08% and −1.60% ± 0.71%, respectively, *p* = 0.03 for both) No significant diastolic dysfunction
Bokslag et al., 2018 [61]	56/131	170 ± 28/157 ± 5	46.5 ± 2.3/44.0 ± 5.6	Assessment of PE-induced predisposition to HFpEF	US	Left ventricular wall thickness was increased in the PE group (0.73 ± 0.11 cm vs. 0.79 ± 0.12 cm, *p* = 0.001 for septal wall thickness, and 0.74 ± 0.11 cm vs. 0.80 ± 0.12 cm, *p* = 0.003 for posterior wall thickness) Left ventricular mass was increased in the PE group (60.5 ± 13.1 g/m^2^ vs. 65.4 ± 14.7 g/m^2^, *p* = 0.035) E/e’ was increased in the PE group (6.86 ± 1.16 vs. 7.86 ± 1.95, *p* < 0.001)
Breetveld et al., 2018 [71]	37/67	100 (79–119)/64 (53–77)	40 (47–43)/36 (33–39)	Assessment of endothelial function and asymptomatic structural cardiac alterations	US	No significant remaining alterations in ventricular structure and function, myocardial strain, and diastolic function
Ciftci et al., 2014 [72]	27/40	60/60	36.44 ± 10.45/33.75 ± 7.95	Assessment of impaired myocardial function and arterial stiffness	US	No significant remaining alterations in ventricular structure and function
Clemmensen et al., 2018 [73]	40/53	144 ± 56/150 ± 43	30 ± 5/30 ± 5	Assessment of ventricular structure and function in formerly preeclamptic women	US	No significant remaining alterations in ventricular structure and function Global longitudinal strain was diminished in the PE group (−21% ± 2% vs. −19.5% ± 2.5%, *p* < 0.001) E/e’ was increased in the PE group (7.1 ± 2.0 vs. 7.95 ± 2.85, *p* < 0.01)
deMartelly et al., 2021 [62]	25/21	Not specified	39.72 ± 6.02/35.76 ± 5.62	Assessment of ventricular structure and function in formerly preeclamptic women and the association of activin A to impaired global longitudinal strain	US	Left posterior wall thickness was increased in the PE group (0.80 (0.69–0.88) cm vs. 0.91 (0.84–1.00) cm, *p* = 0.003) Left ventricular mass was increased in the PE group (62.74 (54.82–69.02) g/m^2^ vs. 70.12 (59.83–80.77) g/m^2^, *p* = 0.04) Global longitudinal strain was diminished in the PE group (−21.92% ± 2.70% vs. −18.31% ± 0.68%, *p* = 0.001) E/A was decreased in the PE group (1.65 (1.50–2.10) vs. 1.30 (1.05–1.50), *p* = 0.002)
Ersbøll et al., 2018 [28]	28/28	101 (25–146)/95 (26–143)	39.1 ± 5.3/38.8 ± 5.6	Assessment of the long-term effect of peripartum cardiomyopathy and PE on cardiac function	CMR	No significant remaining alterations in ventricular structure and function
Ersbøll et al., 2021 [74]	28/28	101 (25–146)/95 (26–143)	39.1 ± 5.3/38.8 ± 5.7	Assessment of the relation between biomarkers and cardiac function after PE or peripartum cardiomyopathy	CMR	No significant remaining alterations in ventricular structure and function
Ghi et al., 2014 [64]	18/16	6–12/6–12	31.0 (24–38)/36.5 (17–49)	Assessment of ventricular structure and function in formerly preeclamptic women	US	Left ventricular mass was increased in the PE group (61.25 (51.5–86.9) g/m^2^ vs. 68.55 (51.0–123.8) g/m^2^, *p* = 0.024) No significant diastolic dysfunction
Ghossein-Doha et al., 2013 [75]	8/20	12, 168/12, 168	33 (32–34) at 12 months, 45 (44–47) at 168 months/31 (30–32) at 12 months, 43 (42–45) at 168 months	Assessment of ventricular structure and function in formerly preeclamptic women	US	No significant remaining alterations in ventricular structure and function, myocardial strain, and diastolic function
Ghossein-Doha et al., 2016 [76]	51 PE	Not specified	33 (30–35)/28 (25–33)	Assessment of ventricular structure and function in formerly preeclamptic women according to recurrence of PE	US	Left ventricular mass was lower in women with recurrent PE when compared to non-recurrent PE (28.5 ± 5 g/m^2^ vs. 32 ± 6 g/m^2^, *p* < 0.05)
Ghossein-Doha et al., 2017 [77]	41/107	4–10 years/4–10 years	Not specified, parous	Assessment of hemodynamical involvement in hypertensive pregnancy disorders	US	Prevalence of HF-B was increased in the PE group (1 (2%) vs. 27 (25%), *p* < 0.05) Concentric remodeling was increased in the PE group (1 (2%) vs. 18 (17%), *p* < 0.05)
Guirguis et al., 2015 [78]	27/39	<5/<5	<45/<45	Assessment of PE as a predictor of diastolic dysfunction	US	No significant remaining alterations in ventricular structure and function Diastolic dysfunction was more prevalent in the PE group (17 (44%) vs. 3 (11%), *p* = 0.006)
Kalapathorakos et al., 2020 [14]	8/6	1–3 days, 1 week, 6 months/1–3 days, 1 week, 6 months	29 (20–41)/29 (23–36)	Assessment of ventricular structure and function in formerly preeclamptic women	CMR	Left ventricular mass was increased immediately postpartum (48 (44–57) g/m^2^ vs. 57 (53–68) g/m^2^, *p* = 0.01), but returned to baseline levels after later examinations
Levine et al., 2019 [84]	29/29	7 (6–9) weeks/6 (5–6) weeks	27.8 ± 5.53/30.7 ± 7.32	Assessment of ventricular structure and function in formerly preeclamptic women in the early postpartum period	US	Global longitudinal strain was diminished in the PE group (−15.15 (−17.63–−12.62)% vs. −13.11 (−15.54–−10.76)%, *p* = 0.04) E/A ratio was diminished in the PE group (1.80 (1.29–2.31) vs. 1.45 (1.13–1.77), *p* = 0.006)
Melchiorre et al., 2011 [10]	37/27	12/12 months	78 (29–38)/33 (29–37)	Assessment of ventricular structure and function in formerly preeclamptic women	US	Diastolic dysfunction occurred more frequently in the PE group (17 (45.9%) vs. 10 (12.8%), *p* < 0.001))
Orabona et al., 2017 [79]	60/60	26 ± 7/30 ± 12	36.87 ± 4.28/38.97 ± 3.71	Assessment of ventricular structure and function in formerly preeclamptic women	US	Concentric remodeling without hypertrophy was present in the PE group (46.7% vs. 0%, *p* < 0.05) Reduced LVEF was present in the PE group (13% vs. 0%, *p* < 0.05) E/e’ differed in the PE group, though no value was given (*p* < 0.05)
Orabona et al., 2017 [85]	30/60	2.2 ± 0.6 years/2.4 ± 0.7 years	35 ± 4/35 ± 5	Assessment of ventricular structure and function in formerly preeclamptic women	US	Global radial, circumferential, and longitudinal strain were impaired in the early-onset PE subgroup (23.3% vs. 0%, *p* < 0.05, 33.3% vs. 0%, *p* < 0.05, 53.3% vs. 0%, *p* < 0.05, respectively)
Orabona et al., 2020 [80]	17 controls with fetal growth restriction/134	6–48/6–48	32 ± 3/30 ± 4	Assessment of ventricular structure and function in formerly preeclamptic women	US	No significant remaining alterations in ventricular structure and function, myocardial strain, and diastolic function
Rafik Hamad et al., 2009 [65]	30/35	6–12/6–12	31 ± 4/31 ± 5	Assessment of ventricular structure and function in formerly preeclamptic women	US	Left ventricular mass was increased in the PE group (40 ± 2 g/m^2^ vs. 47 ± 2 g/m^2^, *p* < 0.001) Septal wall thickness and posterior wall thickness were increased in the PE group (0.89 ± 0.02 cm vs. 0.96 ± 0.02 cm, *p* = <0.001 and 0.83 ± 0.03 cm vs. 0.89 ± 0.02 cm, *p* < 0.001, respectively) Relative wall thickness was increased in the PE group (0.37 ± 0.01 vs. 0.38 ± 0.01, *p* < 0.001) E/A ratio was diminished in the PE group (1.75 ± 0.07 vs. 1.58 ± 0.07, *p* < 0.001) Septal and lateral E/e’ was increased in the PE group (6.98 ± 0.42 vs. 9.08 ± 0.40, *p* < 0.001 and 5.23 ± 0.20 vs. 5.86 ± 0.44, *p* < 0.001, respectively)
Reddy et al., 2019 [60]	Systematic review	Review	Review	Assessment of ventricular structure and function in formerly preeclamptic women	US	Women with a history of preeclampsia show a higher ventricular mass (mean difference; 4.25 g/m^2^; 95% CI, 2.08–6.42; *p* = 0.0001) and relative wall thickness (mean difference; 0.03; 95% CI, 0.01–0.05; *p* = 0.02) Women with a history of preeclampsia show evidence of diastolic dysfunction (mean difference; 0.84; 95% CI, 0.41–1.27; *p* = 0.0001 for E/e’, and −0.08; 95% CI, −0.15 to −0.01; *p* = 0.03 for E/A)
Shahul et al., 2018 [66]	25/32	12/12	31.44 ± 4.96/31.50 ± 6.63	Assessment of ventricular structure and function in formerly preeclamptic women and the association with activin A	US	Interventricular septal wall thickness is increased in the PE group (8 (7–9) mm vs. 9 (8–11) mm, *p* = 0.01)Ventricular mass is increased in the PE group (69.33 ± 14.17 g/m^2^ vs. 77.83 ± 14.30 g/m^2^, *p* = 0.03) Global longitudinal strain was impaired in the PE group (−20.48% ± 2.67% vs. −17.76% ± 2.96%, *p* = 0.003)E/A was diminished in the PE group (1.58 (1.36–1.81) vs. 1.35 (1.11–1.58), *p* = 0.03)
Simmons et al., 2002 [81]	44/15	3 ± 1/3 ± 1 months	29 ± 5/32 ± 6	Assessment of ventricular structure and function in formerly preeclamptic women	US	No significant remaining alterations in ventricular structure and function, myocardial strain, and diastolic function
Soma-Pillay et al., 2018 [86]	45/96	12/12 months	27.2 ± 7.14/28.9 ± 6.83	Assessment of ventricular diastolic function in formerly preeclamptic women	US	No significant remaining alterations in ventricular structure and function, myocardial strain, and diastolic function
Spaan et al., 2009 [82]	29/22	276/276	Not specified, no significant difference	Assessment of remote hemodynamics and renal function in formerly preeclamptic women	US	No significant remaining alterations in ventricular structure and function, myocardial strain, and diastolic function
Strobl et al., 2011 [87]	17/14	14.94 ± 1.6 years/15.78 ± 2.2 years	43.9 ± 3.8/43.6 ± 2.9	Assessment of ventricular structure and function in formerly preeclamptic women	US	No significant remaining alterations in ventricular structure and function, myocardial strain, and diastolic function
Valensise et al., 2008 [67]	1119/107	12/12	32 ± 5/33 ± 4	Assessment of ventricular structure and function in formerly preeclamptic women	US	Left ventricular mass was increased in the PE group (26 ± 5 g/m^2^ vs. 39 ± 10 g/m^2^, *p* < 0.05) Septal and posterior wall thickness was increased in the PE group (0.70 ± 0.09 cm vs. 0.82 ± 0.13 cm, *p* < 0.05, and 0.65 ± 0.10 cm vs. 0.78 ± 0.14 cm, *p* < 0.05, respectively) Relative wall thickness was increased in the PE group (0.29 ± 0.04 vs. 0.33 ± 0.05, *p* < 0.05)
Valensise et al., 2016 [68]	147/53	12–18/12–18	34 ± 4/34 ± 4	Assessment of ventricular structure and function in formerly preeclamptic women	US	Left ventricular mass was increased in the PE group (24.8 ± 5.0 g/m^2^ vs. 30.4 ± 6.8 g/m^2^, *p* < 0.05) Relative wall thickness was increased in the PE group (0.28 ± 0.04 vs. 0.33 ± 0.04, *p* < 0.05) E/e’ was increased in the PE group (7.34 ± 2.11 vs. 9.03 ± 3.43, *p* < 0.05)
Yuan et al., 2014 [83]	7/7	16–20/16–20	Not specified, no significant difference	Assessment of ventricular and carotid structure and function in formerly preeclamptic women	US	No significant remaining alterations in ventricular structure and function, myocardial strain, and diastolic function

## 6. Future Recommendations

### 6.1. Recommendations for Researchers

Previous studies demonstrated indicators of ventricular hypertrophy, either increased left ventricular mass and concentric remodeling, in women with a history of preeclampsia. In addition, diastolic dysfunction is often reported in the cardiac sequelae of preeclampsia, but also regarding systolic contractional dysfunction as determined by lowered global peak strain. Future studies could investigate diastolic strain rate as well, as global peak strain is an inherently systolic marker. CMR-based strain measurements also provide the opportunity to assess radial strain, which is known to suffer from poor reproducibility in speckle tracking-based approaches.

Given the prevalence of heart failure with a preserved ejection fraction in women, research on myocardial microvascular function using MRI perfusion in formerly preeclamptic women is urgently warranted. Fibrosis is an underlying cause of cardiac stiffening and diastolic dysfunction, which could be assessed with T_1_ and ECV mapping in future studies. Together this will provide more insight into the underlying mechanisms of post-preeclampsia cardiac changes, which will aid in developing new precision treatment strategies. In addition, these studies may provide novel imaging biomarkers, which can be used for early detection of heart failure in a stage where myocardial changes are still reversible. Along with progressive stiffening and dynamic loss of diastolic function, cardiac fibrosis can be considered detrimental and irreversible for healthy cardiac functioning. Not only the time course of this process after preeclampsia but also its most important risk factors need to be elucidated to improve timely and targeted preventive strategies. As such, in addition to traditional cardiovascular risk factors, both cardiac ultrasound and MRI are complementary mandatory in a cohort of formerly preeclamptic women to illuminate this interrelation, preferably in a combined clinical research setting. These longitudinal findings may support rational choices in determining the most optimal individualized timing in follow-up.

Future research could also investigate whether women with subtle cardiac changes, such as reduced global myocardial strain, can be reversed with treatment or lifestyle coaching and whether treatment or lifestyle intervention can reduce the number of patients that develop symptomatic heart failure.

Automated image analysis, for instance, by using deep learning, could potentially reduce the interobserver variation, enabling the measurement of subtle changes. In addition, radiomics and deep learning may be able to provide additional new imaging markers for risk stratification.

### 6.2. Recommendations for Clinicians

The cardiac sequelae after preeclampsia are reflected as increased left ventricular mass, impaired myocardial strain, and diastolic dysfunction predispose to the development of clinical heart failure later in life. Regular cardiovascular screening to lower CV risk factors and timely diagnosis of hypertension and aberrant cardiac remodeling is considered crucial in a clinical setting. To date, there is not yet enough evidence to propose an optimal frequency in CVRM management. Nonetheless, considering the substantial increased relative risk and the undetected cardiac adjustments to these risk factors, cardiac ultrasound can be viewed upon of relevant additional value. We do not recommend routine CMR in clinical practice after preeclampsia unless clinically indicated.

## Figures and Tables

**Figure 1 biomolecules-12-00415-f001:**
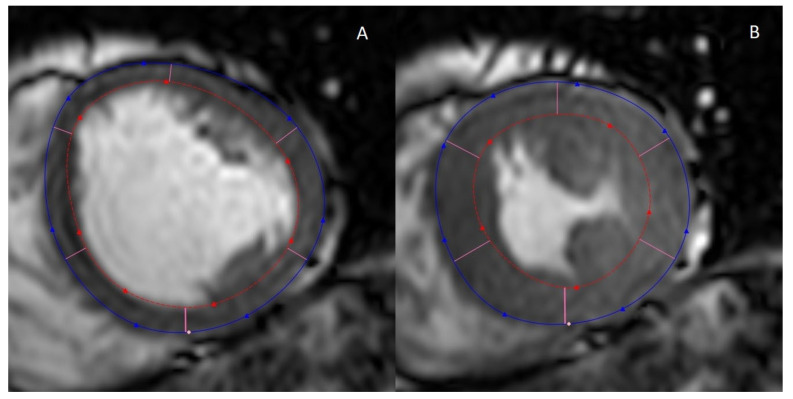
Left ventricular short-axis view of cine MRI. The left ventricle in the (**A**) end-diastolic and (**B**) end-systolic phase. The blue and red lines denote the outer and inner walls of the left ventricle, respectively. Papillary muscles may or may not be included, depending on the individual choice of the researcher/clinician. In this example, they are excluded [21].

**Figure 2 biomolecules-12-00415-f002:**
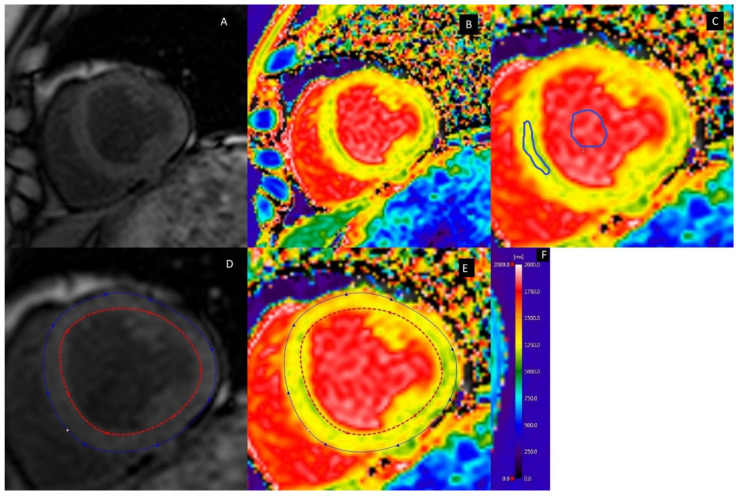
A typical example of a native T_1_ map. (**A**) Anatomical reference image of a left ventricle short-axis view. (**B**) Superimposed heat map showing the different T_1_ relaxation times for different tissues. (**C**) Regions of interest drawn in the mid-septal region and ventricular cavity. (**D**) Global myocardial T_1_ relaxation times can be determined by denotation of the entire left ventricular wall. (**E**) Superimposition of the drawn contours from image D on the T_1_ map. (**F**) Heat map scale with T_1_ relaxation times ranging from 0 to 2000 ms [21].

**Figure 3 biomolecules-12-00415-f003:**
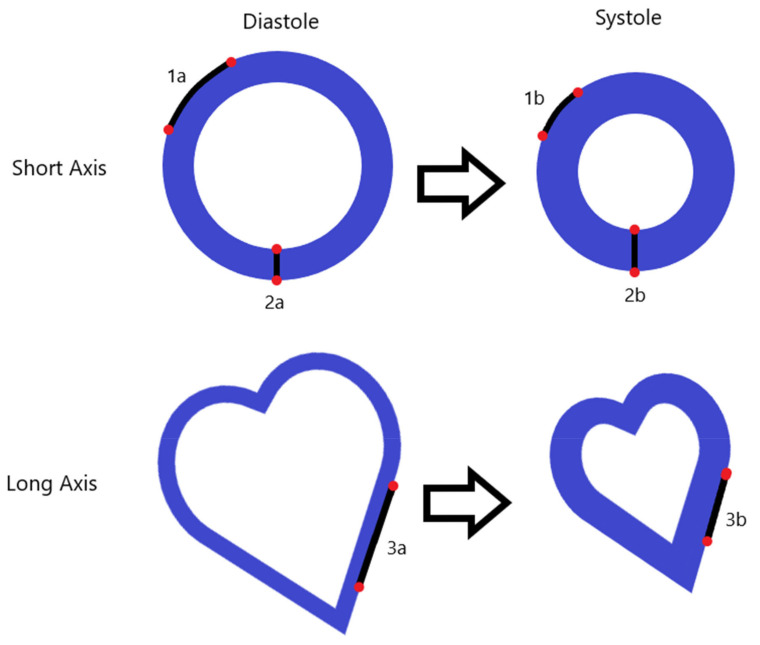
A graphical representation of the principal strain directions. The black lines represent the distance between two hypothetical features (the red dots). 1a and 1b demonstrate circumferential shortening, 2a and 2b demonstrate radial thickening, and 3a and 3b demonstrate longitudinal shortening.

**Figure 4 biomolecules-12-00415-f004:**
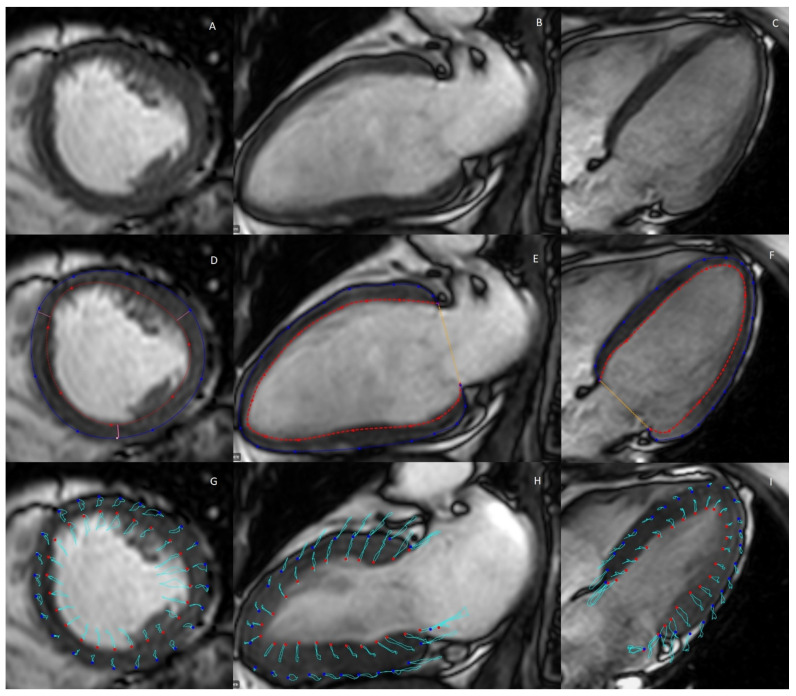
Typical example of calculation of myocardial strain with MRI feature tracking. (**A**–**C**): Anatomical cine MR images of the left ventricular short axis, long-axis 2-chamber, and long-axis 4-chamber view, respectively. (**D**–**F**) Delineation of the inner (red) and outer (blue) left ventricular wall in these image planes. (**G**–**I**) Feature tracking employed over the entire cardiac cycle, with pathing (the motion of myocardial features over time) made visible by the light-blue lines [21].

**Figure 5 biomolecules-12-00415-f005:**
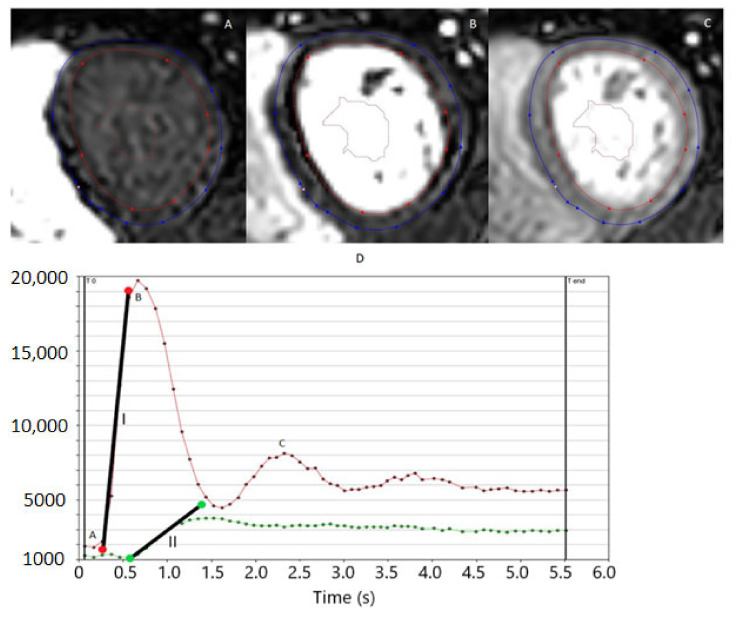
First-pass perfusion. (**A**) shows the contoured mid-height-level myocardium before the first pass of the contrast bolus, though it is already visible in the right ventricle. (**B**) shows the first pass of the bolus as it enters the left ventricle. (**C**) shows the second pass of the bolus, also showing the uptake of contrast agent in the myocardium. (**D**) shows the resulting graphs where signal intensities of the myocardium (green line) and blood pool (red line) are visible. The phases of (**A**–**C**) are also visible on the graph with their corresponding letters. Line I shows the upslope of the blood pool, and line II shows the upslope of the myocardium, from which the relative upslope is calculated. Time is measured in seconds (s).

**Table 1 biomolecules-12-00415-t001:** Benefits and limitations of CMR as opposed to cardiac ultrasonography.

CMR	Cardiac Ultrasonography
Entire left ventricle is depicted	Sectional imaging is usually performed
No ionizing radiation	No ionizing radiation
Suitable spatial resolution (1–2 mm) [57]	Suitable spatial resolution (0.5–2 mm) [57]
Suitable temporal resolution (20–50 ms) [57]	Superior temporal resolution (<5 ms) [57]
Superior soft-tissue contrast	Poor soft-tissue contrast
Use of navigators or self-gating allows for free breathing, breathholds may also be applied	Free breathing is possible, though sometimes breathholds are required
High costs	Lower costs
Bedside scan not possible	Bedside scan is possible
Long scan times	Short scan times
MRI contraindications	No contraindications
Accessibility may vary depending on location	Readily accessible
Results are operator-independent	Results are highly operator-dependent
Enables tissue characterization using T_1_ and ECV mapping, thus allowing for diffuse and focal fibrosis assessment	Unable to perform tissue characterization
Diastolic function is assessable through 4D flow CMR	Diastolic function is readily assessable through Doppler and tissue Doppler ultrasonography
Myocardial perfusion is assessable through contrast-based perfusion CMR	Myocardial perfusion is assessable through contrast-based ultrasonography
Gold standard for the assessment of ventricular volumes and function (cine MRI) and myocardial strain (tagging)	No gold standard status

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
