# Peer review of "Noninvasive Cardiac Imaging in Formerly Preeclamptic Women for Early Detection of Subclinical Myocardial Abnormalities: A 2022 Update"

_biomolecules, 2022, doi:10.3390/biom12030415_

Round 1
Reviewer 1 Report
The manuscript of Brant et al is a review interesting, original, and well organizing. The adopted point of view could help both clinicians and researchers to face the diagnose of future cardiac diseases in preeclamptic women.
I recommend review the point 6. Future recommendations (line 457) and transform this point in a chapter of recommendations to clinicians related with preeclamptic women in order to approach the early following of the cardiac damage induced by preeclampsia. Furthermore, future directions for both clinicians and researchers should be included.
Author Response
We thank the reviewer for the valuable comments. We have added a section on recommendations for clinicians at the end of the manuscript.
Textual changes:
- Added two subsections (lines 463 and 498) titled ‘Recommendations for researchers’ and ‘recommendations for clinicians’ respectively.
- Recommendations for researchers: Lines 480-488: “Along with progressive stiffening and dynamic loss of diastolic function, cardiac fibrosis can be considered detrimental and irreversible for healthy cardiac functioning. Not only the time course of this process after preeclampsia, but also its most important risk factors need to be elucidated to improve timely and targeted preventive strategies. As such, additional to traditional cardiovascular risk factors, both cardiac ultrasound and MRI are complementary mandatory in a cohort of formerly preeclamptic women to illuminate this interrelation, preferably in a combined clinical research setting. These longitudinal findings may support rational choices in determining the most optimal individualized timing in follow up. “
- Recommendations for clinicians: Lines 499-507: “The cardiac sequelae after preeclampsia, reflected as increased left ventricular mass, impaired myocardial strain and diastolic dysfunction predispose to the development of clinical heart failure later in life. Regular cardiovascular screening to lower CV risk factors and timely diagnosis of hypertension and aberrant cardiac remodeling is considered crucial in clinical setting. To date, there is not yet enough evidence to propose an optimal frequency in CVRM management. Nonetheless, considering the substantial increased relative risk and the undetected cardiac adjustments to these risk factors, cardiac ultrasound can be viewed upon of relevant additional value. We do not recommend routine CMR in clinical practice after preeclampsia unless clinically indicated. ”

Reviewer 2 Report
This an excellent update of the literature on an important topic: the long term consequences of preeclampsia on the hear function evaluated by imaging through magnetic resonance and ultrasound approaches. the Table1 is certainly useful to this respect, by comparing the two approaches.
In my mind, the most interesting part is the chapter 5 from the results summarized in Table 2; the other previous chapters are nevertheless necessary introductions to the technologies used; it is a bit boring to read, but I guess that researchers and clinicians focused on specific in vivo approaches will gain knowledge of possible alternatives.
The illustrations are useful and clear enough.
I did not see many typos, except the word count that has been forgotten line 274.
Author Response
We thank the author for the valuable feedback, and are pleased to hear that the reviewer considers our update as excellent and on an important topic. We agree that while the technical background may not be the most exciting part to read, it does provide valuable information about diagnostic alternatives for future research and clinical diagnostic approaches. We have removed the word count from chapter 3.
Textual changes:
- Line 282, removed the word count from the chapter title.

Reviewer 3 Report
The authors reviewed the role of noninvasive cardiac imaging in women with cardiovascular disease who had a history of preeclampsia. This review is well-written and informative for the readers. I have some minor comments to improve the manuscript.
Introduction
It is unclear to me why the authors focus on cardiovascular disease after pre-eclampsia. Please clarify why the authors need to focus on the women with prior preeclampsia.
Lines 46-47
I think the authors may clarify the exact rate of cardiovascular disease (e.g. XX % [preeclampsia] versus XX% [control]). Please also cite the previous studies.
Line 95
Please add the high cost of MRI as a hurdle in CMR. In most developed countries, MRI is expensive and is difficult to perform. Please discuss this matter.
Formula 2
I think (4) is not necessary.
Line 190, line 234, line 471-473, Reference lists
Please unify the type of font.
Line 199
((per example should be (per example?
Figure 5
Please clarify the unit of time. (s) Maybe seconds?
Lines 274
Please delete the words count.
Lines 238-258, lines 403-432
These are run-on paragraphs and are excessively long. Run-on sentences often lead to confusion and misinterpretation.
Line 359
Did the authors use MeSH keywords?
Table 2
This table is hard to see. Please revise for better visualization.
Author Response
We thank the reviewer for the valuable comments.
Introduction
It is unclear to me why the authors focus on cardiovascular disease after pre-eclampsia. Please clarify why the authors need to focus on the women with prior preeclampsia.
We focused on cardiovascular disease after pre-eclampsia because of the 2- to 7-fold increase in probability of developing cardiovascular disease following a pregnancy complicated by preeclampsia. Formerly preeclamptic women have persistent remaining cardiac alterations which do not revert to a pre-pregnancy state.
We have made the following adjust to the manuscript: Lines 63-66, “In 25% to 72% of the cases, these cardiac adaptations persist and do not revert to a pre-pregnancy state postpartum, and a majority of women meet the diagnostic criteria for asymptomatic heart failure preterm. This gives formerly preeclamptic women a higher vulnerability to develop cardiovascular disease later in life.”
Lines 46-47
I think the authors may clarify the exact rate of cardiovascular disease (e.g. XX % [preeclampsia] versus XX% [control]). Please also cite the previous studies.
We have made the following adjustment to the manuscript: Lines 48-53 , “A recent study by Vogel et al estimated the worldwide prevalence of cardiovascular disease in women to be 6,403 cases per 100,000. According to a meta-analysis performed by Wu et al, preeclampsia is associated with 4-fold increase in heart failure (Risk Ratio (RR), 4.19; 95% Confidence Interval (CI), 2.09-8.38) , and a 2-fold increase in coronary heart disease and stroke (RR, 2.21; 95% CI, 1.83-2.66 and RR, 1.81; 95% CI, 1,29-1.55, respectively).”
Line 95
Please add the high cost of MRI as a hurdle in CMR. In most developed countries, MRI is expensive and is difficult to perform. Please discuss this matter.
We have added the following sentences: Lines 110-113: “The application of magnetic resonance imaging is often complicated by high costs and lower availability, most notably in developing countries. In many such cases, cardiac ultrasonography remains the modality of choice due to ease of use, lower costs, and more widespread availability.”
Formula 2
I think (4) is not necessary.
We have noticed these numbers in parentheses in all formulas. They appear to be a result of formatting, and cannot be removed without removing the entire formula. We expect that these numbers in parentheses will not be present in the final publication due to the absence of line numbers in the eventual publication.
Line 190, line 234, line 471-473, Reference lists
Please unify the type of font.
We have checked the main body text and found no font inconsistencies. The reference list has been reformatted to fit the main body text.
Line 199
((per example should be (per example?
The second bracket has been removed.
Figure 5
Please clarify the unit of time. (s) Maybe seconds?
The time is indeed expressed in seconds. We have provided further explanation of the unit of time in the figure caption of figure 5.
Lines 274
Please delete the words count.
The word count has been removed.
Lines 238-258, lines 403-432
These are run-on paragraphs and are excessively long. Run-on sentences often lead to confusion and misinterpretation.
We have revised the mentioned paragraphs to improve readability (lines 251-266 and 411-437).
Line 359
Did the authors use MeSH keywords?
MeSH keywords were not explicitly used, but upon review of our search terms, MRI, echocardiography, magnetic resonance imaging, cardiac function, and preeclampsia were found to adhere to MeSH standards.
Table 2
This table is hard to see. Please revise for better visualization.
We have revised Table 2 for improved readability.

Round 2
Reviewer 3 Report
Tue authors revised the manuscript well.